# Predictors of active ageing among older adults in age-friendly communities in Yanji City, China: A cross-sectional study

Jiawei Jiang[1], Ai Theng Cheong[1*], Shariff Ghazali Sazlina[1], Zarina Haron[2], Shanyu Wu[3], Chenli Liang[3], Qi Jiang[3]

**1** Department of Family Medicine, Faculty of Medicine and Health Sciences, Universiti Putra Malaysia, Serdang, Selangor, Malaysia, **2** Department of Nursing and Rehabilitation, Faculty of Medicine and Health Sciences, Universiti Putra Malaysia, Serdang, Selangor, Malaysia, **3** Department of Nursing, Faculty of Nursing, Yan Bian University, Yan Ji City, Ji Lin Province, China

\* cheaitheng@upm.edu.my

## Abstract

### Background

With the global population ageing rapidly, especially in China, promoting active ageing is crucial for ensuring healthy longevity. However, limited studies have examined the levels and predictors of active ageing at the community level in provincially designated age-friendly communities.

### Methods

A cross-sectional study was conducted from August to November 2024 in two age-friendly communities in Yanji City, China. We invited 553 older adults aged 60 years and above using simple random sampling methods. We collected data through structured face-to-face interviews using validated instruments that measured socio-demographic and physical, environmental, health-related, and social variables. We used multiple linear regression to identify significant predictors of active ageing.

### Results

A total of 513 older adults participated 56.9% were female, and 90.4% were aged 60−79. The mean active ageing score was 100.98 (SD = 16.78). Higher educational attainment (β = 0.138, 95% CI [0.513, 8.736]), higher income levels (β = 0.144, 95% CI [1.265, 10.266]), moderate physical activity levels (β = 0.073, 95% CI [0.004, 0.181]), better cognitive function (β = 0.214, 95% CI [0.522, 1.088]), stronger family support (β = 0.124, 95% CI [0.399, 1.535)], close social connectedness (β = 0.277, 95% CI [0.595, 1.021]), and use of community (β = 0.176, 95% CI [3.597, 9.532]) and cultural facilities (β = 0.116, 95% CI [1.659, 6.583]) three or more times a week were

**Data availability statement:** All relevant data are within the manuscript and its Supporting Information files.

**Funding:** The author(s) received no specific funding for this work.

**Competing interests:** The authors have declared that no competing interests exist.

significantly associated with higher active ageing. Depression had a significant negative impact on active ageing scores ($\beta = -0.170$, 95% CI [$-1.362$, $-0.570$]).

## Conclusion

The findings underscore the need for integrated strategies encompassing environmental design, social support systems, physical activity promotion, and mental health care to foster active and meaningful ageing in age-friendly community settings.

## Introduction

An ageing population poses increasing challenges for public health systems globally, particularly in rapidly ageing countries such as China. By 2030, the population aged 60 and above is expected to increase by 34%, reaching 1.4 billion in China [1]. China is at the forefront of this demographic shift, with its elderly population expected to exceed 30% by 2050 [2]. However, rapid urbanisation and evolving family structures have eroded traditional support systems, exacerbating challenges such as chronic disease prevalence, social isolation, and economic insecurity in older adults [3,4].

In response, the World Health Organisation (WHO) proposed the Active Ageing Framework to address these challenges, emphasising health, social participation, and security to improve older adults' quality of life [5]. This has led to initiatives such as the Global Network for Age-Friendly Cities and Communities, which encourage cities to design an inclusive environment promoting active ageing [6]. Despite these policy efforts, empirical research remains fragmented. Most studies focus either on national policy or individual-level indicators. At the policy level, studies have focused on social inequality and health disparities [7], implementing pension reforms [8], and developing lifelong learning strategies [9], often using international datasets like SHARE (Survey of Health, Ageing and Retirement in Europe) to compare active ageing trends in countries [10]. At the individual level, research has primarily examined physical and cognitive health, mental well-being, social support, and social engagement [11–13]. Other studies have emphasised the role of volunteering, digital literacy, and participation in leisure activities in enhancing active ageing [14–16]. In contrast, neglecting the multilevel interactions that shape ageing outcomes.

To address this complexity, our study adopts the Ecological Model for Health Promotion, which conceptualises health behaviours and outcomes as products of dynamic interactions across multiple levels: intrapersonal (e.g., age, education, health status), interpersonal (e.g., family support, social networks), community (e.g., accessibility of facilities, neighbourhood engagement), organisational (e.g., work environment), and policy level [17]. Few studies applied this model in the context of Chinese age-friendly communities. Applying this model allows for a more integrated understanding of how community environments, social systems, and individual capacities collectively influence active ageing.

In this context, Yanji City presents a valuable case for empirical investigations into the status of active ageing in age-friendly communities. Designated in 2020 as a

pilot city under the *Health and Ageing Friendly Cities Action and Management Planning* initiative, supported by the Asian Development Bank. As of 2023, two communities, Dan Shan (designated in 2022) and Yuan Fa (designated in 2023), have received official recognition as age-friendly communities. Yet, to date, no studies have evaluated the influence of implementing age-friendly policies on the active ageing level of older adults in Yanji. Thus, this study aims to assess the level of active ageing and identify its influencing factors among older adults living in age-friendly communities in Yanji City.

## Methods

### Study design and setting

Our study employed a cross-sectional survey design and was conducted in two age-friendly communities, Danshan and Yuanfa, in Yanji City, China. Data collection was carried out between August and November 2024. To help ensure methodological rigour and transparency in reporting, we adhered to the guidelines outlined in the Strengthening the Reporting of Observational Studies in Epidemiology (STROBE) statement [18].

### Participants and Sampling Procedure

The sampling frame comprised a registry of 3,677 older adults residing in Dan Shan and Yuan Fa age-friendly communities, as recorded by the community administrative office in August 2024. Each eligible resident was assigned a unique identification number ranging from 1 to 3,677. Using SPSS version 26.0, a simple random sampling procedure was performed without replacement to select 553 participants based on the target sample size determined through prior power analysis.

Inclusion criteria were: (1) aged 60 years or older, (2) able to communicate in Mandarin (the primary language of the community), and (3) registered residents of the selected communities. Exclusion criteria included (1) refusal to participate, (2) a clinical diagnosis of dementia, or (3) complete dependence on caregivers for activities of daily living.

### Sample size

To detect the associations between active ageing and potential influencing factors, we conducted an a priori analysis using G-Power (3.1) software to perform multiple linear regression (fixed model, $R^2$ deviation from zero) using the F-test family. The effect size was set to $f^2 = 0.05$, representing a small-to-moderate effect according to Cohen's guidelines [19], as cited in Faul et al. [20]. With $\alpha = 0.05$, $1 - \beta = 0.95$, and 10 predictors, the minimum required sample size was 497 participants. Considering an anticipated response rate of 90%, based on previous studies of active ageing among older adults in China [21,22], the final target sample size was 553 participants.

### Data Collection

The research team obtained the official list of residents (3,677 older adults) from the community's administrative offices. After performing the simple random sampling, the selected individuals were contacted by phone and invited to a face-to-face interview at the community centre. Three researchers, one principal investigator and two others with academic backgrounds in nursing and public health carried out the survey using a standardised protocol. For participants with visual impairments or literacy issues, the researchers read the questions aloud and recorded responses. All questionnaires were reviewed immediately for completeness. Participants received a small token (e.g., a towel or tissue box) as appreciation for their participation in the study. A total of 553 questionnaires were distributed.

### Instrument

Guided by an ecological model of health promotion, we selected variables at the intrapersonal, interpersonal, and community levels that are most directly observable through a cross-sectional study. Data were collected using a structured

questionnaire comprising four sections: intrapersonal level (age, biological sex, educational attainment, marital status, living arrangement, and average monthly household income, and the continuous variables physical activity, cognitive function, and depression), interpersonal level (family support and social connectedness), community level (neighbourhood accessibility and facility utilisation), and active ageing. The instruments used in our study had previously been validated in the Chinese population and demonstrated acceptable reliability.

**Section A: Intrapersonal Level.** This section collected basic demographic information through structured interviews, including age, biological sex, educational attainment, marital status, living arrangement, and average monthly household income, as well as the continuous variables of physical activity, cognitive function, and depression.

Physical activity was evaluated using the Chinese version of the Physical Activity Scale (PARS-3) to assess the intensity, duration, and frequency of physical activity [23]. The exercise amount was calculated using the formula intensity × (duration-1) × frequency, with each item rated from 1 to 5, and the total score ranges from 0 to 100. Levels were classified as low (≤19), moderate (20~42), and high (≥43). The retest reliability was reported as 0.82, indicating good reliability.

Depressive symptoms were assessed using the Geriatric Depression Scale (GDS-15) [24], with dichotomous scoring (1 = yes, 0 = no), and the total score ranged from 0 to 15. A cut-off score ≥ 8 indicated depressive symptoms. Cronbach's α was 0.76 [25].

Cognitive function was assessed using the Mini-Mental State Examination (MMSE) [26], covering orientation, memory, attention, calculation, recall, and language ability, comprising 11 items. Scores range from 0 to 30. Cut-off values for cognitive impairment were: ≤ 17 points for illiterate individuals, ≤ 20 points for primary education, ≤ 24 points for secondary education or above, and up to 26 points for individuals with higher education. The Cronbach's α was 0.894.

**Section B: Interpersonal Level.** This section assessed family support and social connectedness.

Family support was measured using the Family Functioning Scale (Adaptation, Partnership, Growth, Affection, and Resolve, APGAR) [27], adapted and validated in China by Lv Fan [28]. It included five dimensions: family adaptation, family cooperation, family adulthood, family emotionality, and family intimacy. Each was scored on a 3-point scale (never = 0, sometimes = 1, often = 2). A total score (0–10) was categorised into 0–3 indicates severe impairment of family support; 4–6 indicates moderate impairment of family support; and 7–10 indicates good family support. The Cronbach's α was 0.894 [29].

Social connectedness was assessed using the 6-item Lubben Social Network Scale [30], which evaluated perceived social support from family and friends. Each item was scored from 0 to 5 points, with total scores ranging from 0 to 30. A score <12 indicated a state of social isolation. The Cronbach's α was 0.89 [31].

**Section C: Community Level.** Neighbourhood accessibility and facility utilisation patterns were used to assess the community level. Neighbourhood accessibility was assessed using the instrument developed by Jang [32], which evaluated the walking distance to six facilities: markets/supermarkets, healthcare centres, public offices, senior welfare centres, bus stops/subway stations, and community service agencies. Walking time was rated on a 4-point scale (<5 minutes = 4 to >30 minutes = 1) with a total score ranging from 6 to 24 points (Cronbach's α = 0.76).

Facility utilisation was measured using five self-reported items that assessed weekly visit frequency to different facilities, including commercial, community services, recreational, cultural, and religious facilities. Frequency was categorised as low (0–2 visits/week), moderate (3–5 visits/week), or high (≥ 6 visits/week) based on prior studies [33].

**Section D: Active Ageing.** Active ageing was measured using the Chinese version of the Active Ageing Scale (AAS), initially developed by Thanakwang, Isaramalai [34] and revised by Zhang, Zhang [35]. The AAS consists of 36 items with seven dimensions: being self-reliant (8 items), being actively engaged with society (8 items), developing spiritual wisdom (5 items), building up financial security (4 items), maintaining a healthy lifestyle (5 items), engaging in active learning (4 items), and strengthening family ties to ensure care in later life (2 items). Each item is scored on a 4-point Likert scale, from "not at all compliant" scoring 1 to "fully compliant" scoring 4. The total score ranges from 36 to 144, with higher total scores indicating better active ageing. Previous literature has reported high internal consistency (α = 0.91–0.932) [36,37].

## Ethical consideration

Ethical approval was obtained from the Yan Bian University Ethics Committee for Research Involving Human Subjects (Yan's Medical Ethics No.10240). Written informed consent was obtained from all participants before data collection, ensuring voluntary participation and confidentiality.

## Data analysis

All statistical procedures were performed using IBM SPSS Statistics version 26.0 (Released 2019; IBM Crop., Armonk, NY, USA). Active ageing scores were treated as a continuous dependent variable in all analyses. Descriptive statistics were used to summarise sample characteristics. Continuous variables were reported as means and standard deviations (SD). Categorical variables were summarised using frequencies (n) and percentages (%).

Univariate analyses were conducted to examine the associations between independent variables and active ageing scores. One-way analysis of variance (ANOVA) was applied for categorical independent variables, and Pearson's correlation coefficient was used for continuous variables that met the assumption of normal distribution.

Variables that showed statistically significant associations ($p < 0.05$) in univariate analyses were subsequently entered in a multiple linear regression model to identify independent predictors of active ageing. To account for potential confounding effects, the model was adjusted for key socio-demographic covariates, including age, educational attainment, marital status, living arrangement, and average family monthly income, to isolate the independent effects of the significant predictors on active ageing. The assumptions of linear regression, including linearity, normality of residuals, homoscedasticity, and independence, were tested and found to be satisfied. Multicollinearity among predictor variables was assessed using the variance inflation factor (VIF), with a VIF value less than 10 indicating an acceptable level of multicollinearity. All statistical tests were two-tailed, and a p-value $< 0.05$ was considered statistically significant.

## Result

A total of 513 participants were collected, resulting in a response rate of 92.8% (513/553). Table 1 presents the demographic characteristics of participants. Most participants (90.4%) were between 60 and 79 years old, with only 9.6% aged 80 or older. Females accounted for a slightly higher proportion (56.9%) than males (43.1%). In terms of education, more than half (64.6%) had at least a junior high school education. Most participants were married (67.4%) and lived with their spouses (57.3%). Economic disparities were evident, as 19.7% of participants had a monthly household income below 1,000 CNY (≈ 138 USD), while 22.6% earned more than 5,000 CNY (≈690 USD). Religious facilities had the lowest engagement among facilities available in the community, with 87.3% reporting low-frequency usage of religious facilities. The level of active ageing was relatively high, with a mean score of $100.98 \pm 16.78$ on a scale ranging from 36 to 144.

The univariate analysis result showed that younger older adults ($F = 12.740$, $p < 0.001$), those with higher education levels ($F = 19.455$, $p < 0.001$), had married ($F = 8.541$, $p < 0.001$), those living with a spouse ($F = 11.160$, $p < 0.001$) exhibited higher active ageing scores. The higher average monthly family income groups displayed a better level of active ageing ($F = 21.484$, $p < 0.001$). Regarding the frequency of use of facilities, the results show that commercial, community services and cultural facilities are associated with active ageing and exhibit a positive trend, with higher frequency of use corresponding to higher active ageing scores ($p < 0.001$), as shown in Table 2.

The Pearson Correlation Analysis revealed significant relationships among various factors influencing active ageing, as shown in Table 3. Accessibility has a slight positive correlation with active ageing ($r = 0.097$, $p < 0.05$). Physical activity is positively associated with active ageing ($r = 0.246$, $p < 0.01$), while depression shows a negative correlation ($r = −0.278$, $p < 0.01$), indicating that higher depression levels are linked to lower active ageing scores. Cognitive function ($r = 0.431$, $p < 0.01$), family support ($r = 0.329$, $p < 0.01$), and higher social connectedness ($r = 0.417$, $p < 0.01$) also demonstrated strong positive correlations with active ageing.

**Table 1. Characteristics of the older adults observed in this study (*n*= 513).**

| Variables | Categories | Frequency | Percentage |
|---|---|---|---|
| Age (years old) | 60-69 | 229 | 44.6 |
| | 70-79 | 235 | 45.8 |
| | ≥ 80 | 49 | 9.6 |
| Sex | Male | 221 | 43.1 |
| | Female | 292 | 56.9 |
| Education level | Primary education | 93 | 18.1 |
| | Junior high school | 146 | 28.5 |
| | Senior high school | 185 | 36.1 |
| | College/ university | 89 | 17.3 |
| Marital status | Married | 346 | 67.4 |
| | Single | 25 | 4.9 |
| | Widowed or divorced | 142 | 27.7 |
| Living arrangement | Alone | 130 | 25.3 |
| | Living with spouse | 294 | 57.3 |
| | Living with offspring | 68 | 13.3 |
| | Others | 21 | 4.1 |
| Average monthly household income (USD) | <1000 CNY (≈138 USD) | 101 | 19.7 |
| | 1000-3000 CNY (≈138–416 USD) | 152 | 29.6 |
| | 3001-5000 CNY (≈416–690 USD) | 144 | 28.1 |
| | >5000 CNY (≈690 USD) | 116 | 22.6 |
| Commercial facilities' usage frequency | Low frequency | 69 | 13.5 |
| | Moderate | 402 | 78.4 |
| | High frequency | 42 | 8.2 |
| Community service facility's usage frequency | Low frequency | 99 | 19.3 |
| | Moderate | 369 | 71.9 |
| | High frequency | 45 | 8.8 |
| Recreational facilities usage frequency | Low frequency | 130 | 25.3 |
| | Moderate | 329 | 64.1 |
| | High frequency | 54 | 10.5 |
| Cultural facilities usage frequency | Low frequency | 298 | 58.1 |
| | Moderate | 171 | 33.3 |
| | High frequency | 44 | 8.6 |
| Religion facilities usage frequency | Low frequency | 448 | 87.3 |
| | Moderate | 38 | 7.4 |
| | High frequency | 27 | 5.3 |
| Accessibility (Mean±SD) | 18.01±2.74 | | |
| Physical activity (Mean±SD) | 21.10±13.28 | | |
| Cognitive function (Mean±SD) | 23.53±4.45 | | |
| Depression (Mean±SD) | 4.33±2.96 | | |
| Family support (Mean±SD) | 7.67±2.14 | | |
| Social connectedness (Mean±SD) | 14.47±5.76 | | |
| Active ageing (Mean±SD) | 100.98±16.78 | | |

**Table 2. Comparison of active ageing level across socio-demographic groups ($n = 513$).**

| Variables | Categories | Number | Active ageing | | |
|---|---|---|---|---|---|
| | | | $M \pm SD$ | $t$ or $F$ | $p$ |
| Age (years old) | 60-69 | 229 | 104.68 ± 16.01 | 12.740 | <0.001 |
| | 70-79 | 235 | 98.94 ± 17.31 | | |
| | ≥ 80 | 49 | 93.49 ± 13.48 | | |
| Sex | Male | 221 | 100.87 ± 16.39 | 0.017 | 0.896 |
| | Female | 292 | 101.07 ± 17.10 | | |
| Education level | Primary education | 93 | 91.18 ± 17.14 | 19.455 | <0.001 |
| | Junior high school | 146 | 99.89 ± 15.62 | | |
| | Senior high school | 185 | 103.25 ± 16.12 | | |
| | College/ university | 89 | 108.30 ± 14.74 | | |
| Marital status | Married | 346 | 103.08 ± 16.83 | 8.541 | <0.001 |
| | Single | 25 | 96.56 ± 11.15 | | |
| | Widowed or divorced | 142 | 96.65 ± 16.59 | | |
| Living arrangement | Living alone | 130 | 100.24 ± 13.96 | 11.160 | <0.001 |
| | Living with spouse | 294 | 103.62 ± 16.03 | | |
| | Living with offspring | 68 | 91.00 ± 18.72 | | |
| | Others | 21 | 101.00 ± 23.83 | | |
| Average monthly family income yuan (USD) | <1000 CNY (≈138 USD) | 101 | 91.50 ± 16.43 | 21.484 | <0.001 |
| | 1000-3000 CNY (≈138–416 USD) | 152 | 99.84 ± 15.83 | | |
| | 3001-5000 CNY (≈416–690 USD) | 144 | 102.84 ± 16.13 | | |
| | >5000 CNY (≈690 USD) | 116 | 108.43 ± 15.02 | | |
| Commercial facilities' usage frequency | Low frequency | 69 | 97.43 ± 19.02 | 7.822 | <0.001 |
| | Moderate | 402 | 100.66 ± 16.10 | | |
| | High frequency | 42 | 109.95 ± 16.63 | | |
| Community service facility's usage frequency | Low frequency | 99 | 94.20 ± 16.19 | 12.269 | <0.001 |
| | Moderate | 369 | 102.07 ± 16.51 | | |
| | High frequency | 45 | 107.00 ± 16.21 | | |
| Recreational facilities usage frequency | Low frequency | 130 | 101.61 ± 17.75 | 0.293 | 0.746 |
| | Moderate | 329 | 100.57 ± 16.22 | | |
| | High frequency | 54 | 102.02 ± 17.93 | | |
| Cultural facilities usage frequency | Low frequency | 298 | 97.79 ± 16.29 | 14.038 | <0.001 |
| | Moderate | 171 | 104.88 ± 16.94 | | |
| | High frequency | 44 | 107.52 ± 14.51 | | |
| Religious facilities' usage frequency | Low frequency | 448 | 101.03 ± 16.56 | 0.632 | 0.532 |
| | Moderate | 38 | 98.74 ± 19.75 | | |
| | High frequency | 27 | 103.44 ± 16.16 | | |

**Note:** Group comparisons were conducted using an independent samples $t$-test (for variables with two categories) and a one-way analysis of variance (ANOVA) (for variables with three or more categories). $F$ values represent ANOVA test statistics; $t$ values represent $t$-test statistics.

The multiple linear regression in Table 4 identified several significant predictors of active ageing, accounting for 44.0% of the variance. Among the intrapersonal variables, participants with junior high school education showed significantly higher levels of active ageing compared to those with primary education (β = 0.138, 95% CI [0.513, 8.736]). Income level, with those earning >5000 CNY (≈690 USD) had higher active ageing levels compared to those earning less than 1,000 CNY (≈138 USD) (β = 0.144, 95% CI [1.265, 10.266]). For community services usage frequency, moderate (β = 0.176,

**Table 3. Pearson correlations between active ageing and the variables of accessibility, physical activity, depression, cognitive function, family support, and social connectedness (n=513).**

| Variables | r | p-value |
|---|---|---|
| Accessibility | 0.097 | 0.029* |
| Physical activity | 0.246 | <0.001** |
| Depression | −0.278 | <0.001** |
| Cognitive function | 0.431 | <0.001** |
| Family support | 0.329 | <0.001** |
| Social connectedness | 0.417 | <0.001** |

**Note:** r represents the Pearson correlation coefficient. * $p < 0.05$, ** $p < 0.001$

95% CI [3.597, 9.532]) and high frequency utilisation (β = 0.116, 95% CI [2.200, 11.568]) were positively associated with active ageing. The use of cultural facilities also showed positive associations: moderate (β = 0.116, 95% CI [1.659, 6.583]) and high frequencies of utilisation (β = 0.122, 95% CI [3.233, 11.389]) were both significantly associated with higher active ageing level.

Other significant predictors included physical activity (β = 0.073, 95% CI [0.004, 0.181]), which showed a positive association. Cognitive function (β = 0.214, 95% CI [0.522, 1.088]) and family support (β = 0.124, 95% CI [0.399, 1.535) were positively correlated with active ageing. Higher social connectedness scores were linked to higher active ageing levels (β = 0.277, 95% CI [0.595, 1.021]). In contrast, depression showed a negative association with active ageing (β = -0.170, 95% CI [-1.362, -0.570]).

## Discussion

Our study investigated the level of active ageing and its influencing factors among older adults living in age-friendly communities in Yanji City, China. The findings revealed a relatively high level of active ageing, influenced by various factors, including educational attainment, average family monthly income, community service, frequency of usage of cultural facilities, physical activity, depression, cognitive function, family support, and social connectedness.

In our study, the mean active ageing score among older adults living in age-friendly communities was 100.98 (SD = 16.78). This score is higher than those reported among urban migrant older adults (Mean = 95.06, SD = 17.93) [21] and rural older adults (Mean = 70.00, SD = 30.00) [37] in a non-age-friendly setting in China. This finding suggests that age-friendly communities may have a positive impact on active ageing. These communities typically offer more developed physical and social environments, such as improved access to health services, structured community-based programs, and stronger social networks [38]. These features were also identified as significant predictors of higher levels of active ageing in our study. In contrast, the settings for the previous two studies did not participate in an age-friendly community program and, therefore, may have lacked such supportive infrastructure [21,37].

Increasingly, the mean score observed in our study was quite similar to that reported by Jiange Zhang et al. in a community-based survey in Zhengzhou City, China (Mean = 102.40, SD = 19.00) [39]. Zhengzhou is a more economically developed city, where older adults tend to have higher education levels, a factor essential for increased active ageing [40]. Therefore, our findings highlight the potential of age-friendly community initiatives to bridge the gap in active ageing between economically disadvantaged and more developed regions, enabling older adults in smaller cities, such as Yanji City, to achieve comparable levels of active ageing.

Among intrapersonal variables, educational attainment and family income emerged as significant predictors of active ageing in our study. We found that older adults with junior high school education levels find it easier to achieve active ageing than those with primary education. Previous studies have shown that schooling enhances health literacy and

**Table 4. Multiple linear regression analysis of factors of active ageing.**

| Variables | Unstandardised Coefficients | | Standardised Coefficients | t | p | 95.0% Confidence Interval for B | | VIF |
|---|---|---|---|---|---|---|---|---|
| | B | SE | Beta | | | Lower bound | Upper bound | |
| (Constant) | 49.992 | 6.378 | | 7.838 | <0.001 | 37.460 | 62.524 | |
| **Age (years old)** | | | | | | | | |
| 60-69 | Ref | | | | | | | |
| 70-79 | −2.020 | 1.257 | −0.060 | −1.607 | 0.109 | −4.491 | 0.450 | 1.278 |
| ≥ 80 | −3.218 | 2.181 | −0.056 | −1.476 | 0.141 | −7.503 | 1.066 | 1.337 |
| **Educational attainment** | | | | | | | | |
| Primary education | Ref | | | | | | | |
| Junior high school | 5.124 | 1.838 | 0.138 | 2.788 | 0.006* | 0.513 | 8.736 | 2.239 |
| Senior high school | 2.082 | 1.921 | 0.060 | 1.084 | 0.279 | −1.692 | 5.856 | 2.769 |
| College/ university | 3.010 | 2.449 | 0.068 | 1.229 | 0.220 | −1.801 | 7.820 | 2.798 |
| **Marital status** | | | | | | | | |
| Married | Ref | | | | | | | |
| Single | −2.336 | 3.179 | −0.030 | −0.735 | 0.463 | −8.583 | 3.911 | 1.525 |
| Widowed or divorced | 3.075 | 2.259 | 0.082 | 1.361 | 0.174 | −1.364 | 7.513 | 3.325 |
| **Living arrangement** | | | | | | | | |
| Living alone | Ref | | | | | | | |
| Living with spouse | 0.437 | 2.202 | 0.013 | 0.199 | 0.843 | −3.889 | 4.763 | 3.861 |
| Living with offspring | −3.529 | 1.998 | −0.071 | −1.766 | 0.078 | −7.454 | 0.396 | 1.493 |
| Living with others | 1.727 | 3.362 | 0.020 | 0.514 | 0.608 | −4.878 | 8.333 | 1.444 |
| **Average family monthly income (USD)** | | | | | | | | |
| <1000 CNY (≈138 USD) | Ref | | | | | | | |
| 1000-3000 CNY (≈138–416 USD) | 0.305 | 1.798 | 0.008 | 0.170 | 0.865 | −3.227 | 3.838 | 2.194 |
| 3001-5000 CNY (≈416–690 USD) | 3.231 | 1.921 | 0.087 | 1.682 | 0.093 | −0.544 | 7.005 | 2.425 |
| >5000 CNY (≈690 USD) | 5.765 | 2.291 | 0.144 | 2.517 | 0.012* | 1.265 | 10.266 | 2.988 |
| **Commercial facilities usage frequency** | | | | | | | | |
| Low frequency | Ref | | | | | | | |
| Moderate | 1.267 | 1.714 | 0.031 | 0.739 | 0.460 | −2.102 | 4.636 | 1.622 |
| High frequency | 4.487 | 2.577 | 0.073 | 1.741 | 0.082 | −0.577 | 9.550 | 1.625 |
| **Community service usage frequency** | | | | | | | | |
| Low frequency | Ref | | | | | | | |
| Moderate | 6.565 | 1.510 | 0.176 | 4.346 | <0.001* | 3.597 | 9.532 | 1.499 |
| High frequency | 6.884 | 2.384 | 0.116 | 2.888 | 0.004* | 2.200 | 11.568 | 1.480 |
| **Cultural facilities usage frequency** | | | | | | | | |
| Low frequency | Ref | | | | | | | |
| Moderate | 4.121 | 1.253 | 0.116 | 3.289 | 0.001* | 1.659 | 6.583 | 1.136 |
| High frequency | 7.311 | 2.076 | 0.122 | 3.522 | <0.001* | 3.233 | 11.389 | 1.100 |
| **Physical activity** | 0.093 | 0.045 | 0.073 | 2.061 | 0.040* | 0.004 | 0.181 | 1.159 |
| **Depression** | −0.966 | 0.202 | −0.170 | −4.791 | <0.001* | −1.362 | −0.570 | 1.156 |
| **Cognitive function** | 0.805 | 0.144 | 0.214 | 5.587 | <0.001* | 0.522 | 1.088 | 1.336 |
| **Family support** | 0.967 | 0.289 | 0.124 | 3.344 | <0.001* | 0.399 | 1.535 | 1.248 |
| **Social connectedness** | 0.808 | 0.108 | 0.277 | 7.450 | <0.001* | 0.595 | 1.021 | 1.265 |
| **Accessibility** | 0.015 | 0.213 | 0.019 | 0.541 | 0.589 | −0.303 | 0.534 | 1.103 |
| *R2* | **0.468** | | | | | | | |
| **Adjusted *R2*** | **0.440** | | | | | | | |

self-management capabilities, facilitating active ageing [41]. Similarly, a higher average monthly household income had a positive influence on active ageing by enabling older adults to access healthcare and social activities that support their physical, mental, and social well-being [42].

In our study, other intrapersonal level factors that were found to influence active ageing significantly included moderate physical activity levels, better cognitive function, and the absence of negative emotions such as depression. All these factors interact and collectively support active ageing. This was reflected in previous studies, which reported that supportive infrastructure alone in age-friendly environments would not automatically lead to higher levels of physical activity among older adults [43]. For example, a survey by Chaudhury et al. demonstrated that the physical activity levels of older adults were influenced by the physical environment of their neighbourhoods, such as a walkable environment (a community-level factor), as well as interpersonal elements like community cohesion and social support from friends [44]. Similarly, another study in Lithuanian noted that the degree of greenery in a community (a community-level factor) did not necessarily translate to higher levels of physical activity. Instead, intrapersonal factors, such as personal motivation, and interpersonal factors, like social engagement, played critical roles in translating environmental features into positive behavioural outcomes [45]. These findings suggest that while age-friendly communities provide the necessary infrastructure to support physical activity, attention should also be paid to the social participation of older adults to encourage higher levels of physical activity.

Mental health, such as cognitive functioning and depression status, was significantly associated with active ageing. Older adults with better cognitive function tended to achieve higher levels of active ageing. Good cognitive functioning supported independence, enhanced self-care, improved health management, increased participation in social activities, and reduced the incidence of Alzheimer's disease, thereby significantly contributing to active ageing [46,47]. These findings were consistent with a longitudinal study conducted in Singapore, which demonstrated a positive relationship between cognitive ability and active ageing [48]. Depression negatively impacted active ageing, which reduced cognitive functioning, impaired daily activities, and negatively impacted emotional well-being [49,50]. There is a need to incorporate the importance of psychological factors and mental well-being, in addition to physical health, into active ageing policies to provide comprehensive strategies for enhancing active ageing in the country [51].

In our study, interpersonal level factors, including family support, were also positive predictors of active ageing. This result is consistent with studies that suggest family support can act as a protective factor against loneliness, depression, and cognitive decline and can promote empathy [52,53].

Our results identified social connectedness as the most influential predictor of active ageing. Literature has shown that close social ties and meaningful interpersonal relationships were consistently associated with enhanced psychological well-being, increased participation in daily life, and a greater sense of purpose among older adults [54]. In our study, older adults residing in age-friendly communities reported relatively high levels of social connectedness (Mean = 14.47, SD = 5.76), suggesting that they are well-integrated within their social environment. This high degree of connectedness may be attributed to the community-based structures and inclusive programmers embedded in age-friendly communities, such as senior university courses and volunteer initiatives. These platforms provide older adults with regular opportunities for interaction, engagement, and mutual support, thereby reinforcing their sense of belonging and social inclusion. Consistent with the World Health Organisation's perspective, such environments promote access to social participation, and sustained friendships help cultivate social connectedness, which in turn supports active ageing, encourages engagement, and fosters psychological resilience [55].

At the community level, recent studies have increasingly emphasised the role of the physical environment in facilitating active ageing [56,57]. Our study confirms this by identifying key environmental factors within age-friendly communities that influence active ageing, particularly the accessibility and frequency of community infrastructure usage. While much of the existing literature emphasised the benefits of enhancing the external physical environment, our findings underscore the importance of how actively older adults engage with available facilities. Specifically, using community services and cultural

facilities more than three times per week was positively associated with a spontaneous and intentional increase in higher levels of active ageing. These results underscore the importance of community-level infrastructure in promoting health, participation, and overall well-being among older adults. Our findings align with Klicnik et al., who emphasised that accessible and meaningful community services in age-friendly communities enhance social inclusion, interaction, and continuity of daily routines and social roles that contribute to a sense of identity and purpose [58]. Similarly, Hernández Lara and Toney [59] noted that frequent engagement with cultural spaces enriched cultural experiences, facilitated social interaction, and promoted the formation of new relationships — key elements in supporting the social and cognitive aspects of active ageing.

### Strengths and limitations of the study

Our study assessed the level of active ageing among older adults living in a newly established age-friendly community designed to support the implementation of ageing policies. By examining the influence of intrapersonal, interpersonal, and community factors, our findings provide empirical evidence for promoting active ageing through age-friendly community development, particularly in resource-constrained or developing regions. This supports the formulation of targeted and actionable strategies to enhance active ageing within similar contexts.

While this study offers valuable insights into the factors influencing active ageing, several limitations should be acknowledged. First, the cross-sectional design of this study limits the ability to infer causal relationships between the identified predictors and active ageing outcomes. Longitudinal or experimental studies are needed to establish temporal and causal links among these variables. Second, data were collected through self-report measures, which may be subject to recall bias and social desirability bias. Third, the sample was drawn exclusively from two designated age-friendly communities in Yanji City, which may limit the generalisability of the findings. Moreover, no comparison was made with communities of similar demographic and structural characteristics that have not implemented age-friendly policies. Including such control settings in future research could help to better isolate the potential effects of age-friendly initiatives on active ageing outcomes. Finally, although the ecological model guided the exploration of influencing factors, the organisational and policy levels were not examined in this paper. These dimensions may be more appropriately explored through longitudinal or qualitative approaches to provide a deeper understanding of the pathways linking multi-level determinants to active ageing.

## Conclusion

The older adults in these age-friendly communities achieved relatively high levels of active ageing. The findings suggest that a comprehensive approach to active ageing should integrate the physical environment with interventions to strengthen social networks, encourage physical activity, and support psychological well-being, ensuring that older adults can age actively, healthily, and meaningfully within age-friendly communities.

## Supporting information

**S1 File. The data.**
(XLSX)

**S2 File. The questionnaire.**
(DOCX)

**S3 File. Inclusivity in global research questionnaire.**
(DOCX)

## Acknowledgments

Gratefully acknowledge the participants for their involvement in this study.

## Author contributions

**Conceptualization:** Jiawei Jiang, Ai Theng Cheong, Shariff Ghazali Sazlina.

**Data curation:** Jiawei Jiang, Shanyu Wu, Chenli Liang, Qi Jiang.

**Formal analysis:** Jiawei Jiang, Shanyu Wu, Chenli Liang.

**Investigation:** Jiawei Jiang, Chenli Liang, Qi Jiang.

**Methodology:** Jiawei Jiang.

**Supervision:** Ai Theng Cheong, Shariff Ghazali Sazlina.

**Writing – original draft:** Jiawei Jiang.

**Writing – review & editing:** Ai Theng Cheong, Shariff Ghazali Sazlina, Zarina Haron.

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
