## [Decision Letter · Decision Letter 0]

1 Oct 2025

Dear Dr. Cheong,

Thank you for submitting your manuscript to PLOS ONE. After careful consideration, we feel that it has merit but does not fully meet PLOS ONE’s publication criteria as it currently stands. Therefore, we invite you to submit a revised version of the manuscript that addresses the points raised during the review process.

**ACADEMIC EDITOR:**

We look forward to receiving your revised manuscript.

Kind regards,

Elise Rivera

Academic Editor

PLOS ONE

Journal Requirements:

3. Please remove all personal information, ensure that the data shared are in accordance with participant consent, and re-upload a fully anonymized data set.

Additional guidance on preparing raw data for publication can be found in our Data Policy (https://journals.plos.org/plosone/s/data-availability#loc-human-research-participant-data-and-other-sensitive-data) and in the following article: http://www.bmj.com/content/340/bmj.c181.long .

Additional Editor Comments:

This is a well written manuscript that addresses and under-researched topic. The rigor and quality of this research is high. Just some minor points to address:

Methods: did the authors adjust for any covariates in their models? If so, it would be good to make this clearer to the reader. If not, then please explain why this was not the case. 

In the Limitations section, please mention the limitations of the cross-sectional study design and use of self-report measures. 

Reviewers' comments:

Reviewer's Responses to Questions

**Comments to the Author**

1. Is the manuscript technically sound, and do the data support the conclusions?

Reviewer #1: Yes

2. Has the statistical analysis been performed appropriately and rigorously?

Reviewer #1: Yes

3. Have the authors made all data underlying the findings in their manuscript fully available?

Reviewer #1: Yes

4. Is the manuscript presented in an intelligible fashion and written in standard English?

Reviewer #1: Yes

Reviewer #1: This is a timely and rigorously conducted study of active ageing in designated Chinese age-friendly communities in Yanji. While there is recognition that no studies have actually evaluated the influence of implementing age-friendly policies in Yanji, it may have enhanced the significance of overall findings if a parallel assessment was undertaken in communities similar in size to Dan Shan and Yuan Fa but without any overtly implemented age-friendly policies.

**Do you want your identity to be public for this peer review?** For information about this choice, including consent withdrawal, please see our Privacy Policy

Reviewer #1: No

---

## [Author Response · Author response to Decision Letter 1]

13 Oct 2025

Dear Dr Elise Rivera and Reviewer,

Thank you for your valuable comments and suggestions on our manuscript entitled “Predictors of Active Ageing among Older Adults in Age-Friendly Communities in Yanji City, China: A Cross-Sectional Study” (PONE-D-25-35732). We have carefully revised the manuscript in accordance with the editor’s and reviewers' feedback. Below, we provide a point-by-point response.

Editor’s Comments

1. Methods: Did the authors adjust for any covariates in their models?

Response: Our multiple linear regression analysis was adjusted for key socio-demographic covariates. In our initial submission, the regression results table presented only the variables that were statistically significant in the univariate analyses and were subsequently entered into the final model, for the sake of brevity and clarity. However, we understand that this may have confused.

To address this, we have now replaced the results table in the manuscript with the complete model output, which includes all variables that were entered into the multiple linear regression, regardless of their statistical significance. As shown in the updated Table 4 (Page 17, Lines 284), the model was adjusted for covariates, including age, marital status, and living arrangement, in addition to the other significant predictors. This demonstrates that the identified independent predictors (e.g., social connectedness) are essential even after controlling for these potential confounders.

Additionally, we have revised the Statistical Analysis section in the Methods to make this adjustment explicit:

" To account for potential confounding effects, the model was adjusted for key socio-demographic covariates, including age, educational attainment, marital status, living arrangement, and average family monthly income, to isolate the independent effects of the significant predictors on active ageing."( Page 11, Lines:227-229 )

2. Limitations: Please mention the limitations of the cross-sectional study design and use of self-report measures.

Response: We have revised the Limitations section (Page 22, lines 379–382) to acknowledge that the cross-sectional design precludes causal inference, and that reliance on self-report measures may have introduced recall and social desirability bias.

Response to Journal Requirements:

Response: We have carefully reviewed and reformatted the manuscript to ensure it fully complies with PLOS ONE's style requirements.

2. Please include a complete copy of PLOS’s questionnaire on inclusivity in global research in your revised manuscript.

Response: We have completed the PLOS Questionnaire on Inclusivity in Global Research and will upload it as a Supporting Information file named "S3 Inclusivity in global research questionnaire."

3. Please remove all personal information, ensure that the data shared are in accordance with participant consent, and re-upload a fully anonymized data set.

Response: We have carefully reviewed our dataset in accordance with the PLOS ONE Data Policy and the guidance outlined in BMJ (2010) “Preparing raw clinical data for publication: guidance for journal editors, authors, and peer reviewers” (BMJ 2010;340:c181), which is referenced on the official PLOS ONE website. The dataset did not contain personally identifying information. To further minimize the risk of indirect identification, the continuous variable “age” has been recoded into age groups consistent with the categories used in the manuscript. All other demographic variables (sex, education level, marital status, living arrangement, and average monthly family income) were retained as categorical variables to replicate the analyses. The dataset is now anonymized and compliant with participant consent and ethical requirements. A revised version has been re-uploaded as S1_The database anonymized.xlsx.

Response: We have carefully reviewed the reviewer comments and confirm that no specific previously published works were recommended for citation. Therefore, no additional references were required or added in the revised manuscript.

5. Please review your reference list to ensure that it is complete and correct.

Response: We have carefully reviewed our reference list and confirmed that it is complete and correct. All cited works are relevant, and none have been retracted.

Reviewer’s Comment

This is a timely and rigorously conducted study of active ageing in designated Chinese age-friendly communities in Yanji. While there is recognition that no studies have actually evaluated the influence of implementing age-friendly policies in Yanji, it may have enhanced the significance of overall findings if a parallel assessment was undertaken in communities similar in size to Dan Shan and Yuan Fa but without any overtly implemented age-friendly policies.

Response: We greatly appreciate this insightful suggestion. While this comparative design was beyond the scope of the current study, we acknowledge this limitation and have added a statement in the 'Strengths and limitations of the study' section (page 22, lines 384–387). Future studies may consider such comparative approaches to strengthen the evidence.

These revisions have strengthened our manuscript. We sincerely thank you for your thoughtful review and constructive feedback.

Kind regards,

Cheong AiTheng

---

## [Editor Report · Decision Letter 1]

3 Nov 2025

Predictors of Active Ageing among Older Adults in Age-Friendly Communities in Yanji City, China: A Cross-Sectional Study

PONE-D-25-35732R1

Dear Dr. Cheong,

We’re pleased to inform you that your manuscript has been judged scientifically suitable for publication and will be formally accepted for publication once it meets all outstanding technical requirements.

Kind regards,

Elise Rivera

Academic Editor

PLOS ONE
---

## [Editor Report · Acceptance letter]

PONE-D-25-35732R1

PLOS ONE

Dear Dr. Cheong,

I'm pleased to inform you that your manuscript has been deemed suitable for publication in PLOS ONE. Congratulations! Your manuscript is now being handed over to our production team.

Kind regards,

on behalf of

Dr. Elise Rivera

Academic Editor

PLOS ONE